# A Machine Learning Multi-Class Approach for Fall Detection Systems Based on Wearable Sensors with a Study on Sampling Rates Selection [note 1]

**DOI:** 10.3390/s21030938

**Published:** 2021-01-30

**Authors:** Nicolas Zurbuchen, Adriana Wilde, Pascal Bruegger

**Affiliations:** 1Institute of Complex Systems (iCoSys), School of Engineering and Architecture of Fribourg Switzerland, HES-SO University of Applied Sciences and Arts Western Switzerland, 1700 Fribourg, Switzerland; pascal.bruegger@hes-so.ch; 2Centre for Health Technologies (CHT), School of Electronics and Computer Science, University of Southampton, Southampton SO17 1BJ, UK; 3Department of Digital Technologies, Faculty of Business, Law and Digital Technologies, University of Winchester, Winchester SO22 4NR, UK

**Keywords:** fall detection, wearable sensors, sampling rate, data preprocessing, feature extraction, Machine Learning

## Abstract

Falls are dangerous for the elderly, often causing serious injuries especially when the fallen person stays on the ground for a long time without assistance. This paper extends our previous work on the development of a Fall Detection System (FDS) using an inertial measurement unit worn at the waist. Data come from *SisFall*, a publicly available dataset containing records of Activities of Daily Living and falls. We first applied a preprocessing and a feature extraction stage before using five Machine Learning algorithms, allowing us to compare them. Ensemble learning algorithms such as Random Forest and Gradient Boosting have the best performance, with a Sensitivity and Specificity both close to 99%. Our contribution is: a multi-class classification approach for fall detection combined with a study of the effect of the sensors’ sampling rate on the performance of the FDS. Our multi-class classification approach splits the fall into three phases: pre-fall, impact, post-fall. The extension to a multi-class problem is not trivial and we present a well-performing solution. We experimented sampling rates between 1 and 200 Hz. The results show that, while high sampling rates tend to improve performance, a sampling rate of 50 Hz is generally sufficient for an accurate detection.

## 1. Introduction

Falls are one of the leading causes of death among the elderly [1]. Every year, 28% to 35% of the elderly fall at least once and this rate increases with age [2]. Falls can have severe physical, psychological and even social consequences. They can also heavily affect the independent quality of living. They can result in bruises and swellings, as well as fractures and traumas [3]. A significant risk is the *long-lie*. This happens when an elderly person remains on the ground for a long duration without being able to call for help. It is associated with death within the next few months following the accident [4]. It also affects the elderly’s self-confidence who may develop the *fear of falling*’ syndrome. It leads to anxiety when performing Activities of Daily Living (ADLs) and can lead to subsequent falls [1].

Therefore, the elderly must continuously be monitored to ensure their safety. Families organize visits but these can be inconvenient and even insufficient. Hiring caregivers or moving into nursing homes are sometimes not affordable options. Recent progresses in technology have enabled the development of Assisted-Living Systems (ALSs) [5]. They can assist the elderly and provide a safer environment through constant monitoring while relieving caregivers’ workload. However, ALSs create other challenges such as privacy concerns and acceptability issues that need to be addressed [6].

Fall Detection Systems (FDSs) are part of ALSs. Their goals are to identify falls and notify caregivers so that they can intervene as fast as possible. However, fall recognition is challenging from a computational perspective. Falls can be defined as “the rapid changes from the upright/sitting position to the reclining or almost lengthened position, but it is not a controlled movement” [7]. There is a higher acceleration during falls. Another challenge is that falls can happen in innumerable scenarios. They may occur anywhere at any time [3]. Their starting and ending body posture as well as their direction (e.g., forward, backward) may vary [1]. Hence, FDSs must cover the whole living area. Their reliability must be high while minimizing false alarms, all the while respecting the elderly’s privacy.

This paper is an extension of our work accepted at the ICAIIC 2020 [8]. This paper has three research questions:RQ1:What is the difference in performance across various types of Machine Learning (ML) algorithms in a FDS?To answer this, we developed a reliable FDS by the mean of wearable sensors (accelerometer and gyroscope) and ML algorithms. The goal is to compare lazy, eager and ensemble learning algorithms and assess their results. We implemented five algorithms and tested them in the same setup.RQ2:What is the effect of the sensors’ sampling rate on the fall detection?To study this, we analyzed the influence of the sensors’ sampling rate on the detection. We filtered the data in order to reduce the number of samples measured per second. We then experimented on the filtered data with five ML algorithms. This research question extends our previous work [8].RQ3:What is the difference in performance across various types of ML algorithms by adopting a multi-class approach for identifying phases of a fall?We experimented a different fall detection approach where falls are split into three phases. These are: the period before the fall happens (pre-fall), the fall itself (impact) and after the fall happened (post-fall). This research question extends our previous work [8].

The rest of this paper is organized as follows. In Section 2, we discuss existing FDSs and highlight their distinctive features. Section 3 covers the employed methodology. Section 4 presents and discusses the obtained results. Finally, we conclude with a comment on future work in Section 5.

## 2. Related Work

Scientists have employed various approaches to implement FDSs over the past years. They have been classified as presented in Figure 1. Each of them has its strengths and weaknesses. We focus on wearable technologies since we use this approach. Nevertheless, several survey studies [9,10] reported the other methods in more depth.

### 2.1. Choice of Sensors and Sampling Rate

Several types of sensors including accelerometers, gyroscopes, magnetometers and tilt sensors have been used to detect falls. Based on the fall characteristics, most studies, such as [11,12,13,14,15], employed only acceleration measurements. From our literature review, very few studies use a single gyroscope. For example, Bourke and Lyons [16] used a single biaxial gyroscope and measured changes in angular velocity, angular acceleration and body angle. Tang and Ou [17] also reported promising results, using a single six-axis gyroscope. The separate use of these sensors already produced promising results but their combination is even better [18]. Wang et al. [19] employed a heart rate monitor and discovered that the heart rate increases by 22% after a fall in people over 40 years old. This demonstrates that physiological data can be used in such a system. Across the papers reviewed (summarized in Table 1), the sensors’ sampling rate varied within a range from 10 to 1000 Hz. This variation is not small, one having 100 times more samples than the other, seemingly arbitrarily. Fudickar et al. [20] compared the detection results when varying the sampling rates from 50 to 800 Hz. The results obtained with a sampling rate of 50 Hz were as good as the ones with 800 Hz. Other studies show that low sampling rates can offer reasonable results, for example Medrano et al. [21] used data sampled up to 52 Hz. We further investigate this issue in this paper with similarly low sampling rates.

### 2.2. Sensing Position

The sensor placement highly affects the detection performance. Previous studies [14,33,34] demonstrated that better results are achieved when sensors are placed along the longitudinal axis of the body (e.g., head, chest, waist) when compared to other placements (e.g., thigh, wrist). The movement of this axis during a fall is more consistent and steady. However, this requires to wear a dedicated device on uncommon body parts which consequently creates inconveniences. For this reason, other studies [11,28,29] used commodities (e.g., smartphones carried by the thigh, smartwatches worn on the wrist). These usually do not disturb the users since they already wear them. However, people tend to take these devices off when they are at home which makes the FDS useless. Another method is to combine various sensing positions. Özdemir et al. [27] developed a system consisting of six wearable devices that are all used together. The problem is that the elderly already have acceptability issues with one device, let alone six.

### 2.3. Algorithms

There are two categories of algorithms: *threshold-based* and *ML-based*. Threshold algorithms simply define limit values, outside of which, a fall is detected. They have often been sufficient but they tend to produce false alarms especially with fall-like activities such as sitting abruptly [16]. To compensate, these studies [13,22] added simple posture and pattern recognition algorithms that detect changes in body posture and level of activity. This improves the detection’s robustness while keeping a low computational complexity. However, it may still fail during specific falls and ADLs. For example, Sucerquia et al. [31] used a manual threshold-based classification over their dataset *SisFall*, achieving 96% accuracy.

ML algorithms automatically learn patterns based on data, and very commonly include feature extraction. They require more computational power and are complex to optimize but produce improved results. Most of the studies such as [11,13] employed a supervised learning technique. Common algorithms are k-Nearest Neighbor [27], Support Vector Machine [18,27] and Artificial Neural Network [11,27]. Yuwono et al. [15] used unsupervised learning which works with clusters. This is a compelling solution because it does not require labeled data. The state-of-the-art Deep Learning algorithms are increasing in popularity, achieving promising results in various fields. Musci et al. [36] employed Recurrent Neural Networks to detect falls. They used a publicly available dataset (*SisFall*) [37] and reported outperforming the results of the original paper [31]. Casilari et al. [35] employed a Convolutional Neural Networks on several datasets, including *SisFall* [31]. They reported promising results with a Sensitivity and Specificity over 98%.

### 2.4. Classification Strategies

The objective of FDSs is to identify whether a fall happened or not, hence a binary decision. FDSs previously reported in the literature typically follow a binary classification approach, following the intuition that the event of interest is whether the participant has fallen or not. A notable exception to this trend [25] extends this common approach by aiming to differentiate amongst various causes of falls. The study differentiates three causes of falls which are trips, slips and others. Another study [23] used a different approach where the type of fall is identified (amongst the types forward, backward, lateral) as well as various ADLs. In a different context, which is Fall Prevention System [38], the goal is to detect if a fall will definitively happen in order to deploy a protection mechanism such as airbags. In such systems, it is not the fall that needs to be detected but what we could call the pre-fall, meaning what happens before the actual fall. More recent approaches combine these two ideas, for example [32] used a multi-phase model. They differentiate phases of a fall and then classify them into three classes: free fall, impact and rest phases. We further investigate this, using several ML algorithms as detailed in Section 3.4.

### 2.5. Strengths and Weaknesses

Wearable technologies have several advantages. They are relatively inexpensive and can operate anywhere all of it with minimal intrusion compared to other approaches, such as environmental monitoring [33,34]. In addition, their somewhat limited computational power can be easily overcome with the use of their telecommunication capabilities, which allow the transfer of data for processing outside the device. Wearables can also identify the wearer and get precise measurements. However, they may create discomfort due to their size and intrusiveness. The main disadvantage is their human dependency. These sensors must have enough battery and be worn to work properly. Furthermore, the elderly may have a cognitive impairment and thus, may forget to wear the sensor.

## 3. Materials and Methods

Our FDS is based on a common pipeline (Figure 2) which has been seen in the literature [27]. This pipeline is a common practice when working with ML algorithms. We first acquire raw data using various sensors and convert them into discrete values. We then preprocess the raw data to remove measuring errors which can badly affect the performance. Afterwards, we construct and extract meaningful information in a vector. Finally, we train and evaluate our ML algorithm to distinguish falls from ADLs.

The steps presented above for our FDS pipeline are common to most of our research questions. However, in order to address research question 3, we have a few changes that will be highlighted. Thus, this section is organised as follows: Section 3.1, Section 3.2, Section 3.3, Section 3.4 and Section 3.5 details each step of the pipeline which are common to all research questions. These five subsections answers entirely the first two research questions. However, the third research question requires additional data preparation which is described in Section 3.6.

### 3.1. Dataset

We decided to use a publicly available dataset rather than creating our own experiment with diverse subjects, for reproducibilty purposes. Therefore, in order to select such dataset, we considered those evaluated in a recent meta-review [39]. From those, we pre-selected those who were freely available, as listed in Table 2 which describes each dataset characteristics. We decided to use acceleration measures because studies have shown that interesting performances can be achieved with it. Ultimately, we selected the dataset named *SisFall* [31] over others [40,41] because of its high quality. We assessed this quality with various criteria, namely the size of the dataset and the diversity of subjects in terms of age, gender, weight and height, as detailed in Table 2.

We also took into account the number of falls and ADLs performed by each subject. An additional factor was the sensors’ sampling rate which needed to be high in order to experiment using various sampling rates. In the *SisFall* dataset, two tri-axial accelerometers (ADXL345 and MMA8451Q) and a tri-axial gyroscope (ITG3200) were used at a sampling rate of 200 Hz. These sensors were attached to the waist, following the longitudinal axis, of the subjects in the data collection phase [31]. This location has been proven to be a reliable one from the literature, as discussed in Section 2.2.

We decided not to use the data of the second accelerometer (MMA8451Q) because usual setups only have a single accelerometer. Having decided to use only data from one accelerometer, we chose that with the highest sensing range and the lowest power consumption which seems adequate for the application. Future work could explore whether there is a significant difference between these sensors.

Twenty-three young people (19 to 30 years old) performed 15 types of falls and 19 types of ADLs including fall-like activities. Fifteen elderly people (60 to 75 years old) also performed the same ADLs for more authenticity. There were five trials per activity except for the walking and jogging activities, each of which had only one trial (See Table 3). Hence, *SisFall* contains a total of 4505 records including 2707 ADLs and 1798 falls, making it unbalanced. A total of 38 people including 19 women and 19 men participated. Table 3 lists the falls and ADLs and their duration.

### 3.2. Data Preprocessing

The *SisFall* dataset required minimal preprocessing. We started by equalizing the duration of each record, by equally cutting (*top and tail* in equal measure) reducing the length to 10 s. We chose 10 s to remove any outliers induced by the fall experiment, whilst preserving the fall within each record. To generate various sensors’ sampling rates, we reduced the number of samples in each record. Thus, for a sampling rate of 100 Hz, we removed 50% of the sample along the record.

Regarding the two walking and two jogging activities, which only have one trial (Table 3), we extracted 5 times 10 s for each record. We did this to have the same number of trials per activity. We selected 5 windows with no overlap along each record as follows:From 5 to 15 s.From 25 to 35 s.From 45 to 55 s.From 65 to 75 s.From 85 to 95 s.

The additional data preprocessing required for research question 3 is described separately, in Section 3.6.

### 3.3. Feature Extraction

We then extracted meaningful information from the preprocessed data. This process helps extracting information that better characterize each activity. A common practice, when working with time series, is to extract time and frequency domain features [11,27,28]. In addition to the axes’ features, we calculated the *magnitude* of acceleration and rotation measures, to improve the robustness of the fall detection (e.g., in case of fall-like activities involving fast movements). Thus, we also extracted time-domain features such as the *variance*, *standard deviation*, *mean*, *median*, *maximum*, *minimum*, *delta*, *25th centile* and *75th centile*. Additionally, we extracted frequency-domain features, using a Fast Fourier Transform and we extracted two features: the *power spectral density* and the *power spectral entropy*.

The feature extraction process is as follows. Firstly, various formulae are applied to each record. In our case, each record has a length of 10 s with a number of samples varying from 10 to 2000 depending on the sampling rate. We then selected a sensor axis and used all samples to extract the wanted feature class.

This process was repeated for each of the other sensor axes (3 axes, 2 sensors). We appended each calculation to a vector to characterize the record (Table 4). We applied this process also for each sensor *magnitude*, resulting in a feature vector of 88 features per record (11 feature classes × 8 axes). The resulting vector uniquely defines each activity. The algorithm compares and tries to find patterns using these features in order to correctly classify each activity. For example, a fall would most likely have a large delta on its vertical sensing axis, since a fall is usually defined by a high vertical acceleration.

Finally, we normalized the extracted features to rescale the data to a common scale. This gives more influence to data with small values which can be neglected depending on the employed algorithm. In this work, we used the common *min-max* normalization which scales the values between 0 and 1 included.

### 3.4. Classification Algorithms

We selected 5 different ML algorithms: k-Nearest Neighbor (KNN), Support Vector Machine (SVM), Decision Tree (DT), Random Forest (RF) and Gradient Boosting (GB). These are described in Section 3.4.1, Section 3.4.2, Section 3.4.3, Section 3.4.4 and Section 3.4.5, and implemented in Python using *Scikit-Learn*. We used the default parameters value of the different classifiers from the version 0.23.2 of Scikit-Learn. It is a tool with a simple interface, built on scientific libraries such as *NumPy*, *SciPy*, and *matplotlib*. The library code is open source and is under the BSD license. Moreover, its documentation is very complete and includes many sample codes. We used the default parameters of each classifier of the version

Pedregosa et al. [42] introduced *Scikit-Learn* and presented its features, comparing the efficiency of its algorithms to other similar libraries. The results show that it is often faster and has the advantage of supporting many available algorithms. This led to its wide adoption in the ML community. In particular, *Scikit-Learn* provides several classification and regression algorithms for supervised learning. Moreover, it implements model selection and evaluation functions that allow to perform cross-validations, searches and comparisons with various metrics.

#### 3.4.1. k-Nearest Neighbor (KNN)

KNN is a well-known algorithm with a very simple operating principle. Data are classified, by a majority vote, with the class most represented among its k-closest neighbors. This algorithm belongs to the lazy learning class because it defers the work as long as possible. During the training, it simply organizes data. However, during a prediction, it browses the recorded data to count the classes of its k-nearest neighbors. Therefore, all calculation costs are during a prediction [43].

This algorithm has two main parameters. The first one is the number of neighbors to consider. A big value allows to have a probabilistic information but the estimation locality may be destroyed. Therefore, compromises have to be made and the value of 5 is used typically in the literature. The optimal number of neighbors depends strongly on the type of data. The second parameter is the method to calculate the distance between two data and defines their closeness. The choice of this metric is complicated and the notion of distance depends on the data characteristics [43]. There are several distance formulae but the most commonly used ones are Euclidean, Manhattan and Minkowski.

Throughout our experiments, we confirmed the following characteristics. The advantages of this algorithm are simplicity, efficacy and ease of tuning to find the best hyper-parameters. In addition, the greater the number of training data, the better the performance, which is however still sensitive to noise. Data normalization can then solve this problem. As a disadvantage, this algorithm is sensitive to the curse of dimensionality. The increase in the number of features tends to improve the results but only up to a certain threshold. When this one is reached, the addition of new features degrades the results. This is because irrelevant features influence negatively in the calculation of the distance. Finally, KNN is expensive in memory and in computation in comparison to other algorithms, as corroborated by the literature [27,30].

#### 3.4.2. Support Vector Machines (SVM)

SVM is also a well-known algorithm. It can be employed in supervised and unsupervised learning. It tries to find the best hyperplane which maximizes the margins between each class. When a linear classification is not feasible, SVM can use a technique named *kernel trick* that maps inputs into a higher dimension [43]. It is an eager learning algorithm because it creates a classification model based on the data during the training. When a prediction is asked, it uses the model to determine the class.

SVM has several hyper-parameters affecting the classification results. The most relevant ones are:*C* makes a compromise between the number of misclassified instances and the margins width of the hyperplane. The lower the value, the larger the margins but potentially increasing the number of errors. When the margins are thin, the number of misclassified samples is low, but this can lead to overfitting.*Kernel* changes the employed mathematical function which creates the hyperplane. A typically used kernel is the Radial Basis Function.*Gamma* defines the influence that one data has compared to the other ones. The higher the value, the bigger its influence range, but this can lead to overfitting. With a low value, the model is at risk of underfitting.

SVM has the advantage of being able to find a unique and global solution which comes from the fact that the optimization problem is convex [43]. Thanks to the *kernel trick*, it can produce good results even with a high features space. However, SVM requires greater processing power during the training to find the best hyperplane and also during the predictions to calculate the support vector for each new data, as corroborated by the literature [18,24,27,30].

#### 3.4.3. Decision Tree (DT)

Trees are well known data structures which are used in many different problems. They are applicable in ML and their objective is to create a DT based on the features of each data. Every node of the tree is divided to satisfy the most data until there are only leaves at the end. Therefore, it is an eager learning algorithm because it tries to build the best DT during the training phase [43].

Most of the hyper-parameters are useful to decide when a node must be divided and when the DT must stop. The most relevant ones are:*Criterion* is the function allowing to measure the quality of the split of a node. A commonly used criterion is *Gini impurity*.*Splitter* is the split selection method of each node because there may be several split solutions. A commonly used splitter is the *best split*.*Max depth* defines the maximum depth that a tree can reach during its creation. A big depth complicates the structure and tends to create overfitting on the data. But on the contrary, a low depth tends to create underfitting.*Min samples split* is the minimum number of data required to enable the split of a node. This value is usually low because the higher it is, the more constrained the model becomes, which creates underfitting.*Min samples leaf* is the minimum number of data required to consider a node as a leaf. Its effect is similar to the previous parameter because a high value would create underfitting.*Max features* corresponds to the maximum number of features to take into account when the algorithm searches for the best split. This hyper-parameter depends on the employed data but also tends to produce overfitting when its value is high.

Advantages of DTs are their ease of understanding and interpretation for humans, as it can be visualized [43]. It also requires few data preparations and has a low cost during a prediction because its complexity is logarithmic. However, a tree can become very complex and not generalize enough the data which then produces overfitting. In the same way, an unbalanced dataset will create biased trees. Despite this shortcoming, DTs (J48 in particular) are commonly used in the literature [24].

#### 3.4.4. Random Forest (RF)

RF is an improvement to DTs because it includes many of them as its name *forest* suggests. Its principle is to create multiple trees and train them on random subsets of data. During a prediction, every tree processes the data and the obtained results are then merged to determine the most likely class by a vote [43]. This method is called *bagging*. This algorithm allows to remove the overfitting problem created by DTs. It is part of ensemble learning algorithms whose concept is to combine several ML algorithms to achieve better performance.

The available hyper-parameters are the same as the ones in DTs in addition to one which allows to define the number of trees to use in the forest. A value of 1 is equivalent to the DT algorithm. A high value will usually give better results. However, this creates a high cost in computational power and memory because each tree has to be stored.

One of the strongest advantages of RF is that it can automatically create a list with the most discriminative features. It has also the ability to create confidence intervals which indicate the certainty rate of a predicted class for each data. Its disadvantage is that the ease of interpretation of DTs is lost. This algorithm has also been used in the fall detection literature [24].

#### 3.4.5. Gradient Boosting (GB)

GB is very similar to RF because it also employs multiple trees but in a different manner. The trees do not work in parallel as in RF but sequentially. The output of each tree is used as input of the following one. The idea is that each tree learns iteratively on the errors made by its predecessor. This is called *boosting* [43]. Because GB is composed of DTs, most of the parameters are the same. However, it has additional ones which are:*Loss* defines the loss function which must be optimized.*Learning rate* slows the learning speed of the algorithm by reducing the contribution that each tree produces. This avoids to rapidly create overfitting.*Estimator* corresponds to the number of sequential DTs. A high number would produce good results but a number too high may create an overfitting issue and use more computational power and memory. The idea is to make a compromise between the number of estimators and the learning rate.*Subsample* defines the data fraction used to train each tree. When the fraction is smaller than 1, the model becomes a Stochastic GB algorithm which reduces the variance but increases the bias.

An advantage of this algorithm is that it can produce better results than RF but it potentially has overfitting issues. It also allows to reduce the variance and the bias. However, the model is more complex to create and as a result the training phase is much longer than in other algorithms. Despite this shortcoming, GB is commonly used in the literature [33,34].

### 3.5. Evaluation

The performance evaluation of our FDS under the selected classifiers was done using *k-fold cross-validation*. This required splitting the dataset into *k* sets. k−1 sets are used as training and 1 as testing. The process is repeated *k* times with a different set as the test one. Given that FDSs must be able to detect falls for new people (e.g., unseen data), the test set should not contain people data that the algorithm has been trained on.

We chose a value of k=5. This creates a training set of 80% and a test set of 20%. We filtered the *SisFall* to only keep subjects that performed all activities. Thus, despite being our motivation to develop a FDS for the elderly, we found it necessary to remove the data related to the elderly subjects, as these had not performed simulated falls. Similarly, we removed three young people’s data due to missing records. This leaves us with data from 20 subjects. This number turned out to be ideal as it allowed us to guarantee that no data from a given subject is used for both training and testing (in an 80/20 split). In other words, the trained models would always be tested with data from new subjects. Consequently, we have 1900 ADLs (19 ADLs × 5 trials × 20 subjects) and 1500 falls (15 falls × 5 trials × 20 subjects), resulting in a more balanced dataset of 3400 records.

During the evaluation of ML algorithms, each prediction falls in one of the following categories:*True negative (TN)*: Correct classification of a negative condition, meaning a reject.*False positive (FP)*: Incorrect classification of a negative condition, meaning a false alarm.*False negative (FN)*: Incorrect classification of a positive condition, meaning a missed.*True positive (TP)*: Correct classification of a positive condition, meaning a hit.

Each prediction is added to the count of its category which allows then to calculate various metrics such as the accuracy. A usual representation of these categories is a confusion matrix.

In fall detection, two metrics are especially important: Sensitivity (SE) (Equation (Equation 1)) and the Specificity (SP) (Equation (Equation 2)) [7]. The SE (or *recall*) corresponds to how many relevant elements are actually selected. This is basically the detection probability meaning how many falls have actually been detected. The SP corresponds to how many non-relevant elements are selected, i.e., how many events classified as non-falls are actually non-falls.
(1)$$Sensitivity=\frac{TP}{TP+FN}$$
(2)$$Specificity=\frac{TN}{TN+FP}$$

We also calculated the accuracy (Equation (Equation 3)) and the F1-score (Equation (Equation 4)). Additionally, we calculated the Area Under the Receiver Operating Characteristics Curve (AUROC) as provided in *scikit-learn*. The AUROC is used to evaluate classifiers’ performance which is used in pattern recognition and ML [44]. In simple terms, an AUROC close to the value of one is indicative of a well-performing algorithm, with high true-positive and true-negative rates consistently.
(3)$$Accuracy=\frac{TP+TN}{TP+TN+FP+FN}$$
(4)$$F1-\mathrm{score}=\frac{2\times TP}{2\times TP+FP+FN}$$

### 3.6. Multi-Class Approach Considerations

To answer our third research question, i.e., “What is the difference in performance across various types of ML algorithms by adopting a multi-class approach for identifying phases of a fall?”, we needed to do one more step to prepare the data for the ML algorithms. The goal of this additional step was to divide the fall sample into three parts which are: pre-fall, impact and post-fall. In doing so, two related questions arise: Where should we split the fall sample and what duration should each part have. Given that a fall has been defined as an uncontrolled, high acceleration [7], especially around the impact point, we defined that the latter would be our reference point to split the fall data sample. Based on this definition, we calculated the magnitude of each accelerometer axis along the sample and selected the highest magnitude as the impact point for each sample. The average time between the moment of loss of balance and the impact point is 0.715 s with a standard deviation of 0.1 s [25]. Consequently, we defined the impact part of the fall as a 2 s interval in the sample which includes the impact point, with 1.5 s leading to it, and the remaining 0.5 s after it. This interval is labeled as *impact*. The remaining part of the sample before the impact interval is labeled as *pre-fall* and the remaining, final part is labeled as *post-fall* (note that based on the result of RQ2, we selected a sample frequency of 50 Hz.). Thus, each 10 s fall sample creates three features vector, one for each phase. The impact phase always represents a 2 s window. The remaining 8 s represents the pre- and post-fall phases. Since the magnitude of the fall is not always at the same timestamp, the pre- and post-fall phase duration varies. If the fall happens early in the sample, the pre-fall phase will be much shorter than the post-fall phase. The opposite if the fall happens late in the sample.

To illustrate the above process, we present Figure 3a, a fall sample of the *SisFall* dataset [31]. Each line represents one of the accelerometer’s axis. In it, a peak in the middle is highlighted which is the impact point (shown as a dotted line) (Figure 3b). The dashed lines limit the three parts of the fall, including the 2 s window of the impact interval. The left-hand part is the pre-fall and the right-hand part is the post-fall. The feature extraction step is applied to each phase of the fall as well as the ADLs.

By identifying the three different phases of the fall in the manner described above, the FDS becomes a multi-class problem. More specifically, when ADLs are taken into consideration, it becomes a four-class classification problem. The motivation behind it lies on the importance in differentiating between ADLs and any phases of a fall, as labeled in the *SisFall* dataset. In order to do that, we apply the same ML algorithms as in Section 3.4. As the SVM classifier is a binary classifier, we extended it by choosing a one-vs.-one scheme.

In order to evaluate the performance of such classification, it is possible to use metrics such as SE, SP F1-score and AUROC, presented in Section 3.5. However, these are typically defined for two-class classification and it is important to show how we have extended them for multi-class problems. We evaluated the performance with the same metrics, using the calculation of the *macro* score for SE, SP, F1-score and AUROC. This is the average metric per class which gives the same importance for each class. The other solution is the *micro* score which average the metric by giving more importance to the amount of data per class. As falls happen rarely, it creates unbalanced dataset but it is crucial to detect them correctly, thus the need to give importance to this class. In our multi-class problem, we calculated the SE for a specific class against all the others together as if they were one class. Matches for this specific class represent the positive cases and matches for the combined class represent the negative cases. Applying this step for each class offers four different Sensitivities, which then are averaged using the previously explained macro score, as per Equation (Equation 5). A similar process is applied for SP and F1-score, as shown in Equations (Equation 6) and (Equation 7).
(5)$$SE_{macro}=\frac{1}{\left|Class\right|}\times \sum_{i=1}^{\left|Class\right|}\frac{TP_{i}}{TP_{i}+FN_{i}}$$
(6)$$SP_{macro}=\frac{1}{\left|Class\right|}\times \sum_{i=1}^{\left|Class\right|}\frac{TN_{i}}{TN_{i}+FP_{i}}$$
(7)$$F1-{\mathrm{score}}_{macro}=\frac{1}{\left|Class\right|}\times \sum_{i=1}^{\left|Class\right|}\frac{2\times TP_{i}}{2\times TP_{i}+FP_{i}+FN_{i}}$$

## 4. Results and Discussion

This section presents and discusses the results for each of the research questions listed in Section 1, namely: Section 4.1 presents the comparison of various Machine Learning (ML) algorithms; Section 4.2 talks about the effect of the sensors’ sampling rates on the detection performance, and Section 4.3 presents the results by splitting each fall into its phases.

### 4.1. Fall Detection System (FDS) Performance

Table 5, Table 6, Table 7, Table 8 and Table 9 present the results of the evaluation of our FDS under the selected five ML algorithms, showing that we successfully developed a reliable FDS. The Sensitivity (SE) reached 98.4% and the Specificity (SP), 99.68%, respectively with Gradient Boosting (GB) and k-Nearest Neighbor (KNN). These results outperformed those reported by Sucerquia et al. [31]. From our review of classification algorithms (Section 3.4), we expected ensemble learning algorithms to achieve better performance than the others. In practice, this trend has been confirmed even though there are some exceptions (see Table 6). This is because they use multiple ML algorithms, though the improvement in performance is at the expense of more resources. Support Vector Machine (SVM) had more difficulties to distinguish the activities. However, by tuning some hyper-parameters, its results may improve.

The high quality of these results was unexpected especially without any optimization such as hyper-parameters tuning. We infer that Activities of Daily Living and falls in the *SisFall* dataset are discriminating by default, similar to [16]. Thus, any algorithm can perform very well. However, in real-life conditions, the SE and SP would very likely drop because of the falls heterogeneity as highlighted by Krupitzer et al. [33,34]. The difficulty of obtaining real falls data is the main shortcoming in FDS studies, given that it is challenging to capture them in realistic settings with the elderly, as noted by Bagalà et al. [26], who compiled a database of only 29 real-world falls.

### 4.2. Sensors’ Sampling Rate Effect

Regarding the sensors’ sampling rate, the trend is that the higher the rate the better the results, which is intuitive since more data are considered when creating the feature vector. However, SVM has a different behavior than the other three, as shown in Figure 4. This shows the variation of the different metrics of each algorithm over the sensors’ sampling rate. It peaks with a sensors’ sampling rate of 20 Hz, indicating that the higher sampling rate does not necessarily improve performance. Especially since a high sampling rate comes with disadvantages such as more computational costs and higher battery consumption. Moreover, the results do not suggest that increasing the sampling rate any further would make a meaningful improvement. In our case, the performance no longer increases significantly after reaching 50 Hz. This sampling rate is in fact the typical one used in the reviewed literature, offering the best reported results (Table 1).

### 4.3. Multi-Class Approach Performance

The multi-class approach to identify different phases of falls achieved promising results with an accuracy close to 99% as shown by Figure 5 for two algorithms. The figure presents also the variability of the results over each fold of the cross-validation for each algorithm. The RF and GB algorithms consistently produced good results over the different metrics except for a single fold, which is seen as an outlier in Figure 5a–d. One explanation might be that it is related to data of a subject who performed the ADLs and falls differently to other subjects. The DT algorithm has a the biggest variability across the algorithms followed by KNN. The variability is low, close to 5% from which a high confidence on the algorithms can be inferred. This is the desired behavior for the type of application, where consistency in minimizing both SE and SP is important to facilitate adoption and usefulness of the FDS. Furthermore, the results of this experiment also confirm the expectation about ensemble learning algorithms performance, which had been observed in the results presented in Section 4.1.

Figure 6 presents a deeper insight of the classification results with the confusion matrices of each split of the k-fold cross-validation for the KNN algorithm. The accuracy of this algorithm is the median amongst all algorithms’ accuracies, therefore it is useful to discuss in depth. We can see that the pre-fall and post-fall phases were consistently correctly classified. The main source of misclassifications comes from the other two classes, i.e., ADL and impact. This negative tendency is stronger in the SVM and DT algorithms but lessened in the RF and GB ones. These confusion matrices are interesting because the patterns of misclassifications are consistent to that expected in a binary detection (i.e., ADL vs. fall). Therefore, an approach could involve removing data associated to the correctly-identified phases of pre-fall and post-fall and treat the problem as a binary classification. However, having correctly isolated and identified these phases, these could be used as a supplementary input to confirm the prediction. Suppose for a given sample, a pre-fall and a post-fall are correctly identified, but the impact is predicted as an ADL. Then, by the mean of a threshold on a confidence interval, the misclassified impact could be overridden and corrected. Another solution could simply consider the fact of identifying a pre-fall and a post-fall phase to always raise an alarm for an impact, given the high confidence of the prediction of both phases.

This novel approach usefulness lies on its provision of an added guarantee that the fall is correctly detected, by offering a mechanism to “fix” a potential misclassification. For a given fall sample, the algorithm should identify once each part of a fall, otherwise, it is identified that one or several classifications are incorrect. Additionally, the ability to recognize the pre-fall stage has many useful applications for fall prevention systems, including airbags for example. This could reduce the likelihood of injuries caused by falls.

The obtained results are of high quality in terms of their accuracy, SE, SP, F-1 and AUROC. This may not be the case when applying the system on data collected on the wild, as we identified during the first experiment. As many other datasets in the FDS community, the *SisFall* dataset is highly discriminating between ADLs and falls. Because their samples lack realism, the studies under laboratory conditions will always outperform those in the real world. In particular, from inspecting *SisFall* data, subjects remained still after a fall, but it is unclear if an older person would act in this way during a real fall, particularly if there was no loss of consciousness.

In our experiment, the pre-fall part was very often correctly classified. However, under real conditions, misclassifications may have arisen (for example, as an ADL). This is due to the fact that, in reality, falls are *unexpected* events occurring perhaps in the middle of an ADL. Therefore, the pre-fall phase may be very short, following immediately from the ADL part of the sample. Whereas in the *SisFall* dataset (as shown in Figure 3) the pre-fall part is not an ADL, instead, the subject is “preparing” to fall (i.e., the fall is not unexpected). In addition, the setup of the experiment in the wild will not be the same as in the lab. It would lack the annotation, and therefore the behavior of the algorithm may not be the same (in particular, with regards to dividing samples). With real-life non-annotated data, it is unknown whether the received data is a fall, and hence a sample associated to an ADL would also be divided into various parts. This would require further investigation.

## 5. Conclusions and Future Work

In this paper, we present our development of a Fall Detection System (FDS) using wearable technologies, to investigate and answer the following three research questions:RQ1What is the difference in performance across various types of Machine Learning (ML) algorithms in a FDS?Our FDS implemented several ML algorithms for comparison: k-Nearest Neighbors, Support Vector Machine, Decision Trees, Random Forest and Gradient Boosting. Our results are an improvement over those reported by Musci et al. [36] and Sucerquia et al. [31], with a final Sensitivity and Specificity over 98%. The system is reliable as we were able to test it on a large dataset containing several thousands of Activities of Daily Living (ADLs) and falls. We obtained these results using various ML algorithms which we were able to compare. We observed that ensemble learning algorithms perform better than lazy or eager learning ones. We also further investigated the effect of the sensors’ sampling rate on the detection rate.RQ2What is the effect of the sensors’ sampling rate on the fall detection?We discovered a tendency that a high sampling rate usually produces better results than a lower one. However, it is not necessary to have an extremely high sampling rate (i.e., in the several hundreds). We recommend using a sampling rate of 50 Hz because it produces improved results with any algorithm while keeping a rather low computational cost.RQ3What is the difference in performance across various types of ML algorithms by adopting a multi-class approach for identifying phases of a fall?We found that the multi-class approach to identify the phases of a fall showed promising results with an accuracy close to 99%. In addition, it includes key features which are the possibility for improved performance by adding subsequent logic to the ML algorithm to address possible misclassifications. Given this performance, we would advocate this multi-class approach as being useful in a different contexts such as fall prevention systems.

There is scope for future work. With the high computation resources available nowadays, it would be interesting to explore Deep Learning (DL) algorithms. In our case however, the size of the cleaned dataset is insufficient for this method to be appropriate given the requirements of DL. The much larger OPPORTUNITY dataset [45] for ADLs has been shown as appropriate for the use of the DL methods [46]. There is a study [36] using Recurrent Neural Networks but there are other algorithms available such as Convolutional Neural Networks with the advantage of automatic feature extraction from time series [46]. This reduces the number of steps to implement and removes the question of how many and which features are needed to be extracted. Additionally, it would be very interesting to reproduce the experiment on the sensors’ sampling rate but with DL algorithms. The results may be different from traditional ML algorithms. The *SisFall* dataset allows plenty of experiments. However, the lack of falls data availability in realistic settings is a common challenge in FDS studies, which also affected our study. In particular currently available datasets with falls in realistic settings (such as in [26]) are far too small for ML approaches to be successful, most particularly, for the state-of-the-art DLs.

Further work would benefit from exploring the use of a multi-class approach for FDS using realistic datasets in order to compare against the performance in the lab and further address any misclassification issues arising in that context. The results presented in this work suggest this is worthwhile doing, and the use of such a system shows promise to make a difference in assisting people sustaining falls.

## Figures and Tables

**Figure 1 sensors-21-00938-f001:**
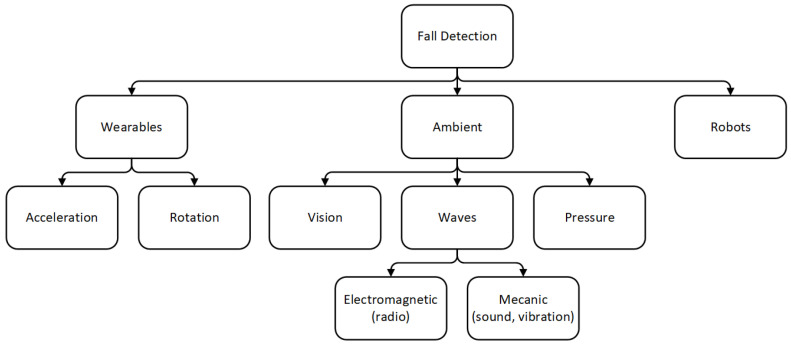
Classification of Fall Detection System approaches.

**Figure 2 sensors-21-00938-f002:**
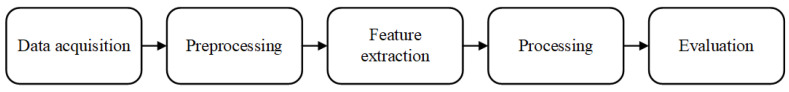
General architecture of Fall Detection Systems.

**Figure 3 sensors-21-00938-f003:**
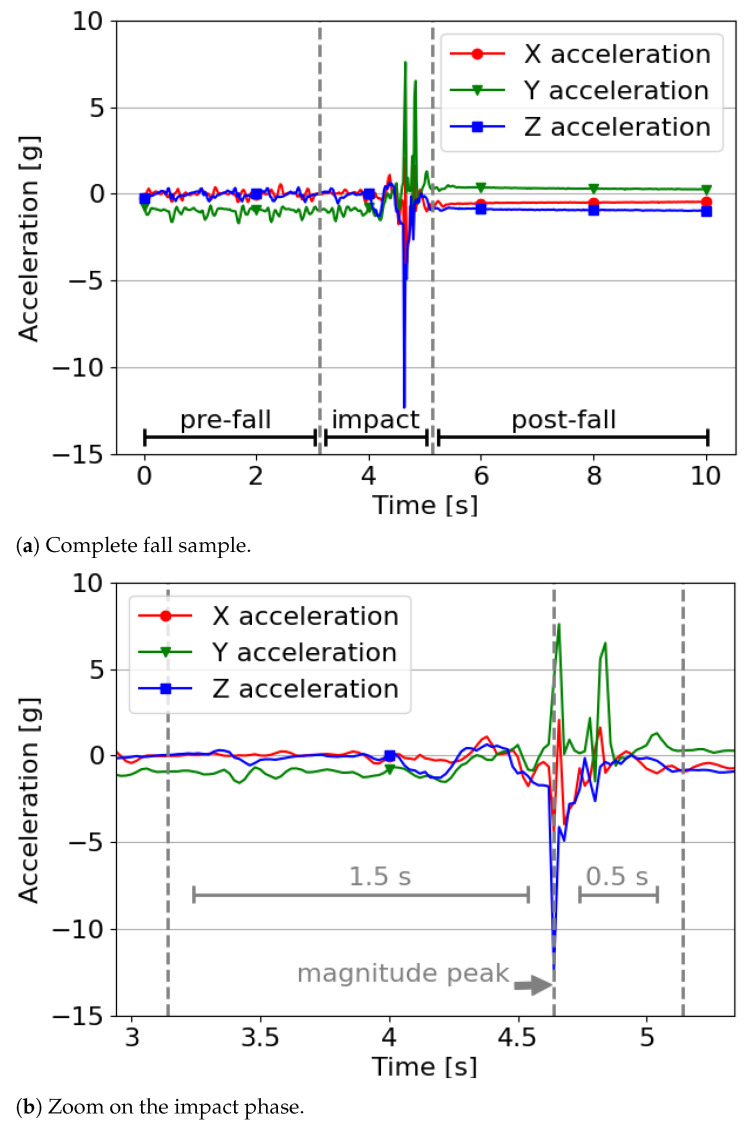
Division of a fall sample into pre-fall, impact and post-fall phases.

**Figure 4 sensors-21-00938-f004:**
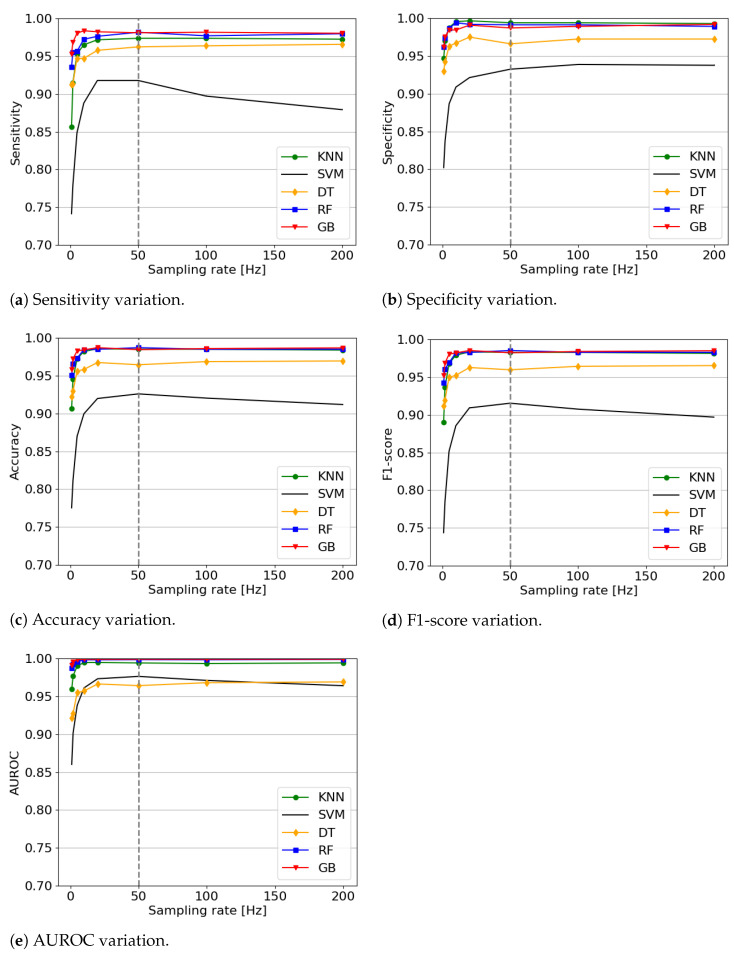
Metrics variation over the sampling rates of five algorithms. The highest average metrics across all algorithms is obtained with a sampling of 50 Hz.

**Figure 5 sensors-21-00938-f005:**
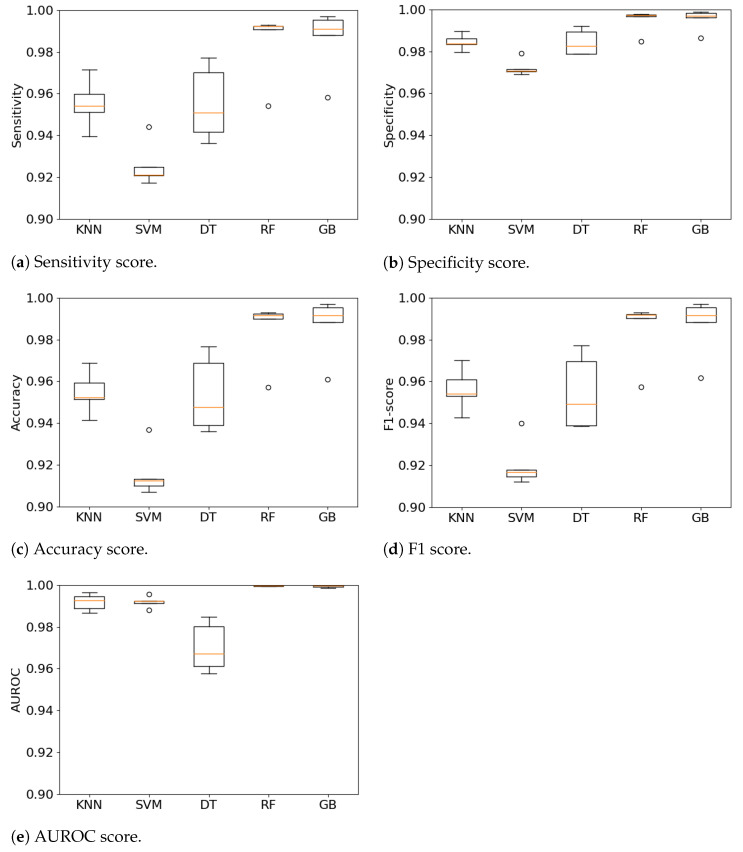
Comparison of various metrics including Sensitivity, Specificity, accuracy, F1 and AUROC of each k-fold split across the ML algorithms.

**Figure 6 sensors-21-00938-f006:**
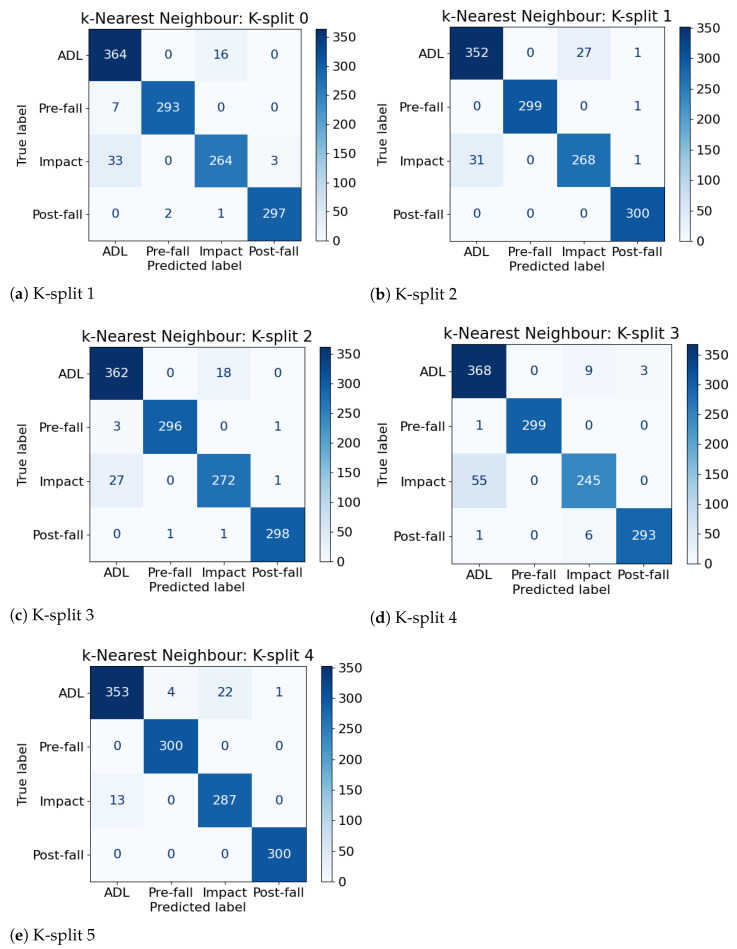
Confusion matrices of the k-Nearest Neighbor Machine Learning algorithm whose accuracy is the median amongst all other algorithms’ accuracies.

**Table 1 sensors-21-00938-t001:** Reviewed studies that used wearable sensors for fall detection (including acronyms at the end of the table).

Research Authors (Year)	Sensors	Freq.	Algorithm	Reported Outcomes
Hwang et al. [22] (2004)	Accelerometer, gyroscope and tilt sensor placed at the chest.	Not reported	Threshold on each sensor compared sequentially.	Accuracy 96.7%
Bourke et al. [12] (2007)	Accelerometer placed at the thigh and chest.	1 kHz	Double acceleration thresholds applied on both sensors.	SP 100%
Bourke et al. [16] (2008)	Bi-axial gyroscope placed at the chest.	1 kHz	Treble angular thresholds.	SE 100% SP 100%
Kangas et al. [14] (2008)	Accelerometer placed at the waist, head and wrist.	400 Hz	Several simple algorithms including thresholds and posture recognition.	SE 97.5% SP 100%
Dinh et al. [18] (2009)	Accelerometer and gyroscope placed at the chest.	40 Hz	Supervised ML algorithms (SVM, Naïve Bayes, C4.5, Ripple-down rules and RBF.	Accuracy 97%
Choi et al. [23] (2011)	Accelerometer and gyroscope placed at the belt.	10–18 Hz	Naive Bayesian Algorithm to identify specific falls and ADLs.	Accuracy 99.4%
Gjoreski et al. [24] (2011)	Four accelerometers placed at the chest, waist, thigh and ankle	6 Hz	Several simple algorithms including thresholds and posture recognition.	Accuracy 99%
Aziz et al. [25] (2011)	Three accelerometers placed at the sternum, right ankle and left ankle	120 Hz	Linear discriminant analysis to identify three causes of fall.	SE 96% SP 98%
Yuwono et al. [15] (2012)	Accelerometer placed at the waist.	20 Hz	Unsupervised ML algorithms (clustering, MLP and augmented RBF neural network) with WT.	SE 100% SP 99.33%
Bagalà et al. [26] (2012)	Accelerometer placed at the lower back.	100 Hz	Comparison of several threshold-based algorithms with posture recognition.	SE 83% SP 94%
Abbate et al. [11] (2012)	Accelerometer from a belt worn smartphone.	50 Hz	Neural network with 8 features extracted as input and a 4 classes classification.	SE 100% SP 100%
Chan et al. [13] (2013)	Three accelerometers placed at the chest.	62.5 Hz	Combination of thresholds, posture measurements and posture recognition.	SE 95.2% SP 100%
Fudickar et al. [20] (2014)	Accelerometer from a smartphone worn at the hip.	50–800 Hz	Threshold-based with sequential posture recognition.	SE 99%
Wang et al. [19] (2014)	Accelerometer and cardiotachometer placed at the chest.	Not reported	Treble thresholds including impact magnitude, trunk angle and heart rate.	SE 96.8% SP 97.5%
Medrano et al. [21] (2014)	Smartphone accelerometer in a pocket (for 95% of ADL data), a hand bag (5%), or two smartphones in separate hand bags (for falls).	unstable, 16.7–52 Hz	One-class SVM, kNN (k = 1), kNN-sum (k = 2) and K-means + 1 NN (k = 800)	SE > 89% SP > 88%
Özdemir et al. [27] (2014)	Accelerometer, gyroscope and magnetometer placed at the head, chest, back, wrist, ankle and thigh.	25 Hz	Features extraction at the total peak acceleration and use of ML algorithms (KNN, LSM, SVM, BDM, DTW and ANN).	SE 100% SP > 99%
Vilarinho et al. [28] (2015)	Accelerometer and gyroscope from the smartphone and smartwatch respectively placed at the thigh and wrist.	Not reported	Acceleration threshold and pattern recognition from both devices	SE 63% SP 78%
Casilari et al. [29] (2015)	Accelerometer and gyroscope from the smartphone and smartwatch respectively placed at the thigh and wrist.	Not reported	Several thresholds compared to each other with every combination of sensors.	SE 96.7% SP 100%
Gibson et al. [30] (2016)	Accelerometer placed at the chest.	50 Hz	Combination of several algorithms (ANN, KNN, RBF, PPCA, LDA)	SE > 90% SP > 90%
Sucerquia et al. [31] (2017)	Accelerometer placed at the waist.	200 Hz	Threshold-based classifier with feature extraction.	Accuracy 96%
Hsieh et al. [32] (2017)	Accelerometer placed at the waist.	128 Hz	Threshold-based Classifier followed by SVM.	Accuracy > 98.74%
Krupitzer et al. [33,34] (2018, 2019)	Accelerometers placed at the chest, waist and thigh.	20–200 Hz	Self-adaptive pervasive fall detection system combining multiple datasets.	SE 75%
Tang et al. [17] (2018)	Six-axis gyroscope inside a bracelet worn at the wrist.	Not reported	Three-feature vector fed into SVM.	Accuracy 100%
Casilari et al. [35] (2020)	Accelerometry signals from several datasets mainly placed at the waist.	10–200 Hz	Deep Learning with Convolutional Neural Networks	SE > 98% SP > 98%

BDM: Bayesian Decision Making; DTW: Dynamic Time Warping; LDA: Linear Discriminant Analysis; LSM: Least Squares Method; PPCA: Probabilistic Principal Component Analysis; WT: Wavelet Transform; ANN: Artificial Neural Network; MLP: Multilayer Perceptron; RBF: Radial Basis Function.

**Table 2 sensors-21-00938-t002:** Main characteristics of the considered datasets. Adapted from [39].

Characteristics	Casilari et al. [40] (2016)	Sucerquia et al. [31] (2017)	Micucci et al. [41] (2017)
Dataset name	*UMAFall*	*SisFall*	*UniMiB SHAR*
No. of sensing points	5	1	1
No. of sensors per point	3	3	1
Type of sensors	*A*|*G*|*M*	*A*|*A*|*G*	*A*
Positions of the points	*Ch*|*Wa*|*Wr*|*Th*|*An*	*Wa*	*Th*
Sampling rates per sensor [Hz]	20|20|20|100|20	200|200|200	50
No. of types of ADL/Falls	12/3	19/15	9/8
No. of samples ADL/Falls)	746 (538/208)	4505 (2707/1798)	7013 (5314/1699)
No. of subjects (Female/Male)	(8/11)	38 (19/19)	30 (24/6)
Subjects’ age range	18–68	19–75	18–60
Subjects’ weight range [kg]	50–93	41.5–102	50–82
Subjects’ height range [cm]	155–195	149–183	160–190

A: Accelerometer; G: Gyroscope; M: Magnetometer; An: Ankle; Ch: Chest; Th: Thigh; Wa: Waist; Wr: Wrist.

**Table 3 sensors-21-00938-t003:** Details of the Activities of Daily Living and falls contained in the *SisFall* dataset [31].

Activity	Duration [s]
Walking slowly	100
Walking quickly	100
Jogging slowly	100
Jogging quickly	100
Walking upstairs and downstairs slowly	25
Walking upstairs and downstairs quickly	25
Slowly sit and get up in a half-height chair	12
Quickly sit and get up in a half-height chair	12
Slowly sit and get up in a low-height chair	12
Quickly sit and get up in a low-height chair	12
Sitting, trying to get up, and collapse into a chair	12
Sitting, lying slowly, wait a moment, and sit again	12
Sitting, lying quickly, wait a moment, and sit again	12
Changing position while lying (back-lateral-back)	12
Standing, slowly bending at knees, and getting up	12
Standing, slowly bending w/o knees, and getting up	12
Standing, get into and get out of a car	25
Stumble while walking	12
Gently jump without falling (to reach a high object)	12
Fall forward while walking, caused by a slip	15
Fall backward while walking, caused by a slip	15
Lateral fall while walking, caused by a slip	15
Fall forward while walking, caused by a trip	15
Fall forward while jogging, caused by a trip	15
Vertical fall while walking, caused by fainting	15
Fall while walking with damping, caused by fainting	15
Fall forward when trying to get up	15
Lateral fall when trying to get up	15
Fall forward when trying to sit down	15
Fall backward when trying to sit down	15
Lateral fall when trying to sit down	15
Fall forward while sitting, caused by fainting	15
Fall backward while sitting, caused by fainting	15
Lateral fall while sitting, caused by fainting	15

**Table 4 sensors-21-00938-t004:** List of extracted time and frequency domain features.

Feature Classes	Domain
Variance	Time
Standard deviation	Time
Mean	Time
Median	Time
Maximum	Time
Minimum	Time
Delta (peak-to-peak)	Time
25th Centile	Time
75th Centile	Time
Power Spectral Density	Frequency
Power Spectral Entropy	Frequency

**Table 5 sensors-21-00938-t005:** Comparison of the Sensitivity across the ML algorithms, with the highest values in bold.

Frequency [Hz]	KNN [%]	SVM [%]	DT [%]	RF [%]	GB [%]
1	85.66	74.13	91.26	93.60	95.33
2	91.46	77.93	91.33	95.53	96.86
5	95.33	84.86	94.73	95.66	98.06
10	96.53	88.80	94.73	97.26	**98.40**
20	97.20	91.80	95.80	97.66	98.26
50	**97.40**	91.80	96.26	**98.20**	98.13
100	**97.40**	**93.89**	96.40	97.73	98.20
200	97.26	93.78	**96.60**	98.00	98.06

**Table 6 sensors-21-00938-t006:** Comparison of the Specificity across the ML algorithms, with the highest values in bold.

Frequency [Hz]	KNN [%]	SVM [%]	DT [%]	RF [%]	GB [%]
1	94.68	80.21	93.00	96.21	96.21
2	97.05	83.73	94.26	97.42	97.57
5	98.78	88.68	96.32	98.68	98.47
10	99.57	90.89	96.73	99.42	98.47
20	**99.68**	92.15	**97.52**	99.21	99.10
50	99.42	93.26	96.63	99.15	98.73
100	99.42	**93.89**	97.26	99.15	98.94
200	99.31	93.78	97.26	98.94	**99.21**

**Table 7 sensors-21-00938-t007:** Comparison of the accuracy across the ML algorithms, with the highest values in bold.

Frequency [Hz]	KNN [%]	SVM [%]	DT [%]	RF [%]	GB [%]
1	90.70	77.52	92.23	95.05	95.82
2	94.58	81.17	92.97	96.58	97.26
5	97.26	87.00	95.61	97.35	98.29
10	98.23	89.97	95.85	98.47	98.44
20	**98.58**	92.00	96.76	98.52	**98.73**
50	98.52	**92.61**	96.47	**98.73**	98.47
100	98.52	92.05	96.88	98.52	98.61
200	98.41	91.20	**96.97**	98.52	98.70

**Table 8 sensors-21-00938-t008:** Comparison of the F1-score across the ML algorithms, with the highest values in bold.

Frequency [Hz]	KNN [%]	SVM [%]	DT [%]	RF [%]	GB [%]
1	88.98	74.35	91.18	94.28	95.24
2	93.68	78.47	91.97	96.08	96.88
5	96.81	85.17	94.99	96.94	98.06
10	97.93	88.56	95.25	98.23	98.23
20	**98.36**	90.93	96.29	98.30	**98.55**
50	98.30	**91.55**	95.99	**98.55**	98.25
100	98.30	90.76	96.45	98.31	98.42
200	98.17	89.70	**96.55**	98.31	98.52

**Table 9 sensors-21-00938-t009:** Comparison of the AUROC across the ML algorithms, with the highest values in bold.

Frequency [Hz]	KNN [%]	SVM [%]	DT [%]	RF [%]	GB [%]
1	95.97	86.02	92.13	98.72	99.12
2	97.73	90.17	92.79	99.26	99.61
5	99.03	93.83	95.52	99.60	99.87
10	99.49	96.14	95.73	99.85	**99.93**
20	**99.50**	97.35	96.66	99.85	99.92
50	99.44	**97.66**	96.44	99.87	**99.93**
100	99.36	97.13	96.83	99.86	**99.93**
200	99.45	96.43	**96.93**	**99.90**	**99.93**

## Data Availability

Not applicable.

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
