# Peer review of "A Machine Learning Multi-Class Approach for Fall Detection Systems Based on Wearable Sensors with a Study on Sampling Rates Selection†"

_sensors, 2021, doi:10.3390/s21030938_

Round 1

Reviewer 1 Report

The article presents a study on fall detection using IMU signals in a public dataset. The main contributions are related to the comparison of several Machine Learning algorithm (not very original), a study on the effect of sampling rate and a study about the improvement of the detector using a multi-class approach.

The paper needs improvement on several points before it can be published.

1) The related work section must be improved:

1.1 There are other papers that study the effect of sampling rate. Please include some of them and discuss if your results are similar to previous studies, for instance:

- Fudickar et al., Threshold-based Fall Detection on Smart Phones . In Proceedings of the International Conference on Health Informatics - Volume 1: HEALTHINF, (BIOSTEC 2014) ISBN 978-989-758-010-9, pages 303-309. DOI: 10.5220/0004795803030309

- Medrano et al., Detecting Falls as Novelties in Acceleration Patterns Acquired with Smartphones, PlosOne, 2014, https://doi.org/10.1371/journal.pone.0094811

1.2 There are other papers that explicitly include several phases in the fall (not only the kind of fall or whether a fall will happen as explained by the authors in section 2.4). For instance:

Hsieh et al., Novel Hierarchical Fall Detection Algorithm Using a Multiphase Fall Model, Sensors, 2017, doi:10.3390/s17020307

1.3 Deep Learning is considered to be the state of the art classifiers. There have been several studies on fall detection using this kind of techniques. The topic deserves more info in the "Related Work" section, for instance:

- Casilari et al.: A Study on the Application of Convolutional Neural Networks to Fall Detection Evaluated with Multiple Public Datasets, Sensors, 2020, DOI:10.3390/s20051466

2) The explanation of the extension to a multi-class problem is not trivial and it is not well explained. In fact this should be the main contribution of the article in my opinion. You should expand the paragraph of lines 378-386 to include more details. For instance: how are the scores defined for a binary classification problem extended to the multi-class case? What is a Positive or a Negative in each case? Does each 10 s window of a fall sample become a sample for Pre-fall, a sample for impact and a sample for post-fall? How was the SVM extended to the multi-class problem (one-vs-one or one-vs-rest)?

Please use equations for performance parameters to make everything clear.

--------------------

There are also some minor points:

3) When talking about scikit-learn, line 217: "The results show that it is often faster but has the advantage of supporting many available algorithms."

This is just a minor detail. The statement sounds strange. The word "but" it is usually used for contrary ideas. However, both aspects, being faster and supporting many algorithms, are positive. I think the sentence should be re-written.

4) A typo: Line 234: Minowski should read Minkowski.

5) Which are the parameters of the models? If your code is not publicly available, at least give all the parameters used for SVM (C, gamma), DT (Max depth, Min samples, etc), RF (number of trees), and GB (loss function, learning rate, etc).

6) Lines 406-408. For future research, you can have a look at the FARSEEING project. I think they give some real fall data under demand (see Bagala et al, Evaluation of Accelerometer-Based Fall Detection Algorithms on Real-World Falls, PlosOne, 2012, doi:10.1371/journal.pone.0037062)

In this regard, the post-fall might not be realistic. It seems from figure 3 that the subjects were told to be still after the fall, but it is unclear if an older person would act in this way. May be you should comment this on the paper.

7) The influence of frequency shown in figure 4 refers only to sensitivity. Do the other performance parameters show the same trend? Please explain this in the article.

8) How did you get the AUROC? Did you use some function of scikit-learn?

Reviewer 2 Report

The paper analyses the performance of several Machine Learning algorithms when aplied to  Fall Detection Systems (FDS) to discriminate between falls and ADLs. The effect of some parameters, such as the sampling rate, is also analysed and discussed.

The paper is well-written, the related works are adequatelly described and compared to the author's proposal. The results are clearly presented, and the conclussions are supported by them. So, I think the paper is suitable for its publication in sensors. 

Author Response

Thank you for the comments.

Round 2

Reviewer 1 Report

The authors have answered all my previous questions.

I have just two points:

  • Line 527: Parentheses are not closed.
  • In figure 3, revise the units of the acceleration. They are supposed to be m/s2 according to your figure. However, in ADL or in the post-fall phase in which there seems to be no movement, the true acceleration norm should be not far from 1 g = 9.81 m/s2 (or exactly 1g if volunteer is still). However, in the figure the acceleration norm is far less than 9.81 m/s2 in those situations.

Author Response

We answered your points in such manner:

  • Line 527: Parentheses are not closed
    --> We closed the parenthesis.
  • In figure 3, revise the units of the acceleration. They are supposed to be m/s2 according to your figure. However, in ADL or in the post-fall phase in which there seems to be no movement, the true acceleration norm should be not far from 1 g = 9.81 m/s2 (or exactly 1g if volunteer is still). However, in the figure the acceleration norm is far less than 9.81 m/s2 in those situations.
    --> We changed the unit from [m/s2] to [g].

Many thanks for the attention to detail.

This manuscript is a resubmission of an earlier submission. The following is a list of the peer review reports and author responses from that submission.

Round 1

Reviewer 1 Report

The authors developed a relatively reliable fall detection system based on the wearable sensors and different machine learning algorithms. The influence of the sampling rate on the accurate detection was analyzed. This work is interesting and useful. Some suggestions are presented below.

(1) More detailed explanation on the k-Nearest Neighbor, support vector machine, and decision tree should be provided.

(2) How to determine a fall from the feature parameters? More detailed explanation should be provided.

(3) How to extract the feature parameters by the algorithm?

Reviewer 2 Report

2) Related Work:

- "only acceleration measurements. Only Bourke and Lyons [15] used a single biaxial gyroscope" (p.2, line 59) -> more works use several sensors, e.g.:

- different frequencies: table with the frequencies used in studies with overviews on used algorithms and reported metrics/results

- "and outperformed the paper’s results." (p.3, line 95/96) -> this paper's results? Which paper?

- In general: rather short and very superficial: What are the results? Which strategies are superior?

- 2.4 strengths and weaknesses -> I would expect a discussion of the existing approaches, not the characteristics of wearables. A clear motivation for the research gap is missing so far. Are there any studies that discuss the different performances for various frequencies?

- "require little computational power" (p.3, line 99) -> In the context of Deep Learning? Please be more concise about that (computations are done on intermediary devices or in the cloud)

3)

- "We decided not to use the data of the second accelerometer (MMA8451Q) because usual setups only have a single accelerometer." (page 4, line 124/125) -> I agree, that you do not target sensor fusion. However, the argument for not using the data of the second sensor as a dedicated device for comparison is not convincing.

- 3.4:  Decision Trees, Random Forest and XG-Boost has been used in other studies, too. k-NN and SVM had been used by many studies, not only 21. Please add furher references, e.g., cf.:
--- H. Gjoreski, M. Lustrek, and M. Gams, “Accelerometer placement for posture recognition and fall detection,” Proc. IE, pp. 47–54, 2011. (all but XG-Boost)
--- Christian Krupitzer, Timo Sztyler, Janick Edinger, Martin Breitbach, Heiner Stuckenschmidt, and Christian Becker. Beyond position-awareness -- extending a self-adaptive fall detection system. Pervasive and Mobile Computing, 58, 2019 (XG-Boost)

3.5:
- Why k=5 and not K=10 as common standard?
- "Thus, we removed all data created by elderly people because they have not performed simulated falls" (p. 7, line 193/194) -> I can understand the reasons for removing elderly people. However, it contradicts the introduction that motivates the work from a point of view of elderly people.
- "we have 3400 records consisting of 1900 ADLs and 1500 falls, making the whole data more balanced" (p. 7, line 196/197) -> this might be more balanced, but what about overfitting as falls are now over-represented?

Metrics:
- I think the F1/F_alpha score is another important measurement as it considers both the precision and the recall (=sensitivity).
- For completeness: How are the accuracy and the AUROC calculated?

4: Results
- "In general, ensemble learning algorithms achieved better performance than the three others. This is because they use multiple ML algorithms, though the improvement in performance is at the expense of more resource" (p. 8, line 216-218) -> Please provide references. Further: You did not analyze the performance, how can you definitely be sure that ensemble learning requires more resources?
- " We could then use a threshold to improve the SE, even though it would reduce the SP and raise more false alarms." (p. 8, line 221/222) -> How does this work?
- How did you implement the algorithms? Which framework has been used for the algorithms?
- You mentioned that related work showed an influence of the position on the performance of the algorithms. How does this influence your results?

5: Conclusion:
- "We observed that ensemble learning algorithms perform better than lazy or eager learning ones." (p.9, line 243/244) -> might be true but has not been discussed in Sect. 4. In general, the discussion of the results is really short.
- "We recommend using a sampling rate of 50 Hz because it produces improved results with any algorithm while keeping a rather low computational cost." (p. 9, line 247-249) -> You only show this for one data set. I would like to see a comparison with other data sets. Further: Computational costs are not evaluated, hence, you cannot provide statements about them!
- I am not an expert, but AFAIK Deep Learning requires a lot of data. Not sure, if the SisFall data set after your data cleansing provide enough data for that.

Minor:
- Fig. 3 - different algorithms are hard to distinguish
- Providing the results in 5 tables (one for each algorithm) make them hard to compare

Reviewer 3 Report

The authors propose a Fall Detection System using an accelerometer combined with a gyroscope 6 worn at the waist. Data come from a publicly available dataset containing records of Activities of Daily Living and falls. Authors  used preprocessing and a feature extraction and then they  applied 5 Machine Learning algorithms, and  compare them. The main contribution is the study of the effect of the sensors’ sampling rate on the performance of the Fall Detection System.

This paper presents work in an important field and to the best of my knowledge, the paper is original and unpublished. The paper is well written and well organized.

In section 2, the study of the literature could be more recent, more recent articles are available on fall detection.

In section 3, the authors describe their system starting from a common pipeline of fall detection system, the proposed methodology and a block diagram is shown in fig.2.

In the section 4, results are presented. I suggest to add colors to the graph in figure 3 to better distinguish the results.

Round 2

Reviewer 2 Report

After reading the former conference paper, I have to change my mind regarding this paper. As contributions, you mention: 

1.) Developing a reliable FDS by the mean of wearable sensors (accelerometer and gyroscope) and various Machine Learning (ML) algorithms. The goal is to compare lazy, eager and ensemble learning algorithms and assess their results. We implemented five algorithms and tested them in the same setup.
2.) Analyzing the influence of the sensors’ sampling rate on the detection. We filtered the data in order to reduce the number of samples measured per second. We then experimented on the filtered data with five ML algorithms.

1 is the main contribution of the conference paper. This is not clearly mentioned. Most of Section 1-3.4 is copy-pasted from the conference paper. I do not really see a clear extension from the previous conference paper. The second contribution alone does in my opinion not satisfy a new journal paper as it is more or less a variation of one evaluation parameter.